# The Prognostic Value of Pre-Treatment Circulating Biomarkers of Systemic Inflammation (CRP, dNLR, YKL-40, and IL-6) in Vulnerable Older Patients with Metastatic Colorectal Cancer Receiving Palliative Chemotherapy—The Randomized NORDIC9-Study

**DOI:** 10.3390/jcm11195603

**Published:** 2022-09-23

**Authors:** Gabor Liposits, Halla Skuladottir, Jesper Ryg, Stine Brændegaard Winther, Sören Möller, Eva Hofsli, Carl-Henrik Shah, Laurids Østergaard Poulsen, Åke Berglund, Camilla Qvortrup, Pia Osterlund, Julia S. Johansen, Bengt Glimelius, Halfdan Sorbye, Per Pfeiffer

**Affiliations:** 1Department of Oncology, Odense University Hospital, 5000 Odense, Denmark; 2Department of Clinical Research, University of Southern Denmark, 5000 Odense, Denmark; 3Academy of Geriatric Cancer Research (AgeCare), 5000 Odense, Denmark; 4Department of Oncology, Regional Hospital Gødstrup, 7400 Herning, Denmark; 5Department of Geriatric Medicine, Odense University Hospital, 5000 Odense, Denmark; 6OPEN—Open Patient Data Explorative Network, Odense University Hospital, 5000 Odense, Denmark; 7Department of Oncology, Trondheim University Hospital, 7030 Trondheim, Norway; 8Department of Clinical and Molecular Medicine, Norwegian University of Science and Technology, 7030 Trondheim, Norway; 9Theme Cancer, Karolinska University Hospital, 171 76 Stockholm, Sweden; 10Department of Oncology, Aalborg University Hospital, 9000 Aalborg, Denmark; 11Department of Immunology, Genetics and Pathology, Uppsala University, 751 05 Uppsala, Sweden; 12Department of Oncology, Tampere University Hospital and Tampere University, 33100 Tampere, Finland; 13Department of Oncology, Helsinki University Hospital, 00014 Helsinki, Finland; 14Karolinska Institute and Karolinska University Hospital, 171 76 Stockholm, Sweden; 15Department of Medicine, Copenhagen University Hospital, 2730 Herlev, Denmark; 16Department of Oncology, Copenhagen University Hospital, 2730 Herlev, Denmark; 17Department of Clinical Medicine, Faculty of Health and Medical Sciences, University of Copenhagen, 2200 Copenhagen, Denmark; 18Department of Oncology, Haukeland University Hospital, 5021 Bergen, Norway; 19Department of Clinical Science, University of Bergen, 5021 Bergen, Norway

**Keywords:** older patients, colorectal cancer, frailty, biomarker, C-reactive protein, neutrophil-to-lymphocyte ratio, Interleukin-6, YKL-40, chemotherapy, survival

## Abstract

Appropriate patient selection for palliative chemotherapy is crucial in patients with metastatic colorectal cancer (mCRC). We investigated the prognostic value of C-reactive protein (CRP), derived neutrophil-to-lymphocyte ratio (dNLR), Interleukin (IL)-6, and YKL-40 on progression-free survival (PFS) and overall survival (OS) in the NORDIC9 cohort. The randomized NORDIC9-study included patients ≥70 years with mCRC not candidates for standard full-dose combination chemotherapy. Participants received either full-dose S1 (Teysuno) or a dose-reduced S1 plus oxaliplatin. Blood samples were collected at baseline and biomarkers were dichotomized according to standard cut-offs. Multivariable analyses adjusted for age, sex, ECOG performance status, and treatment allocation; furthermore, C-statistics were estimated. In total, 160 patients with a median age of 78 years (IQR: 76–81) were included between 2015 and 2017. All investigated biomarkers were significantly elevated in patients with either weight loss, ≥3 metastatic sites, or primary tumor in situ. In multivariable analyses, all markers showed significant association with OS; the highest HR was observed for CRP (HR = 3.40, 95%CI: 2.20–5.26, *p* < 0.001). Regarding PFS, statistically significant differences were found for CRP and IL-6, but not for dNLR and YKL-40. Applying C-statistics, CRP indicated a good prognostic model for OS (AUC = 0.72, 95%CI: 0.67–0.76). CRP is an easily available biomarker, which may support therapeutic decision-making in vulnerable older patients with mCRC.

## 1. Introduction

Colorectal cancer (CRC) is the second most common malignancy worldwide and both its incidence and prevalence peak in adults aged 70 years or older emphasizing the role of aging in CRC pathogenesis [1,2]. Aging is the strongest non-modifiable risk factor for developing cancer [2] and as the general population is aging globally, the number of older adults is going to increase; including those with CRC [3].

Despite the largest proportion of patients with CRC is ≥70 years, older patients are under-represented in randomized controlled trials (RCTs) and are treated based on data extrapolated from younger and fit cohorts [4,5]. However, evidence obtained in younger and fit cohorts cannot be generalized; older patients often benefit less and experience more toxicity of the same treatment affecting their survival and quality of life (QoL) negatively [5] due to comorbidities, impaired organ function, and geriatric syndromes [6].

Although direct evidence obtained in vulnerable older patients with metastatic CRC (mCRC) has been desired for a long time, only a few prospective RCTs investigated this area, and the optimal approach remains unclear [7,8,9]. The American Society of Clinical Oncology and the International Society of Geriatric Oncology recommend the implementation of geriatric assessment (GA) in order to identify the vulnerable older patients who will likely benefit of anti-cancer treatment [10,11]. However, GA is still not available in most oncology practices; alternatively, easily available biomarkers might be an option to optimize patient selection, given that blood sampling is routine procedure in oncology [12].

The investigator-initiated randomized phase II NORDIC9-study compared two commonly applied clinical approaches in older vulnerable patients with mCRC: full-dose monotherapy vs. reduced-dose combination chemotherapy. The detailed protocol, survival outcomes, and several secondary endpoints have already been published and showed that reduced-dose combination chemotherapy resulted in significantly improved progression-free survival (PFS), less toxicities and hospital admissions [9,13]. Furthermore, preservation of global QoL and physical functioning were associated with a reduced-dose doublet [14].

The presence of systemic inflammation is associated with adverse outcomes in several types of cancer [15,16,17]. The NORDIC9-study therefore included a planned analysis of biomarkers indicating systemic inflammation: C-reactive protein (CRP), the derived neutrophil-to-lymphocyte ratio (dNLR), YKL-40 (also named chitinase-3-like 1 protein, CHI3L1), and interleukin-6 (IL-6).

CRP is an acute phase protein and a non-specific marker of tissue damage and inflammation. High CRP in patients with stage I-III CRC is associated with increased risk for postoperative complications, relapse, and mortality [18,19,20]. In patients with mCRC, high baseline CRP is associated with shorter survival [21]. Moreover, CRP is known as a biomarker of frailty, is linked to sarcopenia, functional decline, increased risk of toxicity, decreased QoL, and higher mortality in older adults with different types of malignancies [15,22,23,24,25,26].

When systemic inflammation is present, total white blood cell (WBC) count tends to be elevated caused by an overweight of absolute neutrophil count (ANC) [27]. Consequently, the neutrophil-to-lymphocyte ratio (NLR) will increase. NLR has a prognostic value in patients with cancer and correlates with frailty in older adults with cancer [16,17]. When lymphocyte count is not measured directly, the derived NLR (dNLR) can be calculated; dNLR = ANC/(WBC-ANC). The dNLR has been widely validated and showed similar prognostic ability as NLR [28,29,30].

YKL-40 plays an important role in tumor genesis and progression [31]. Plasma YKL-40 increases with age in healthy individuals [32] and may reflect the chronic low-grade inflammation (inflammaging). Plasma YKL-40 is also a biomarker of age-related diseases, including cancer, characterized by tissue remodeling and inflammation [32]. High plasma YKL-40 is associated with short survival in patients with different types of cancer [33]; in early CRC, high plasma YKL-40 is correlated to poor outcomes [20,34,35]. In the metastatic setting, the NORDIC VII study showed an association between high plasma YKL-40 and short survival [36].

IL-6 is an inflammatory cytokine contributing to systemic inflammation and the development of several age-related diseases, e.g., cancer [37,38]. High plasma IL-6 affects both cancer specific outcomes (relapse and survival) and patient-centered outcomes as functional decline (performance status, decreased mobility, deficits in instrumental activities of daily living) [15,23,39,40,41,42]. IL-6 is considered a prognostic biomarker also in patients with CRC [18,21].

The aim of the current biomarker analysis was to investigate whether these four biomarkers were associated with overall survival (OS) and PFS in vulnerable older patients with mCRC, hence, they may contribute to optimal patient selection for palliative chemotherapy.

## 2. Patients and Methods

### 2.1. Study Design and Participants

The NORDIC9-study, a prospective randomized phase II multi-center study involved 23 centers in four Nordic countries and included patients ≥70 years with mCRC who were unsuitable for full-dose combination chemotherapy [9,13]. The study was approved in all four countries by the National Ethical Committees and was conducted according to the Declaration of Helsinki and the International Conference of Harmonisation Good Clinical Practice guidance. The NORDIC9-study was registered at EudraCT (reg.no. 2014-000394-39). This manuscript was prepared according to the Consolidated Standards of Reporting Trials (CONSORT) guidelines and fulfilled the criteria of the Reporting Recommendations for Tumor Marker Prognostic Studies (REMARK) checklist [43,44].

### 2.2. Interventions

Patients were randomly assigned (1:1) to treatment with full-dose S1 (Teysuno, Taiho Pharmaceutical Co., Ltd., Tokyo, Japan) monotherapy (30 mg/m^2^ orally twice-daily on days 1–14, every three weeks (q3w)) or with reduced-dose SOx (S1, 20 mg/m^2^ orally twice daily + oxaliplatin 100 mg/m^2^ intravenously on day 1, q3w). The treatment protocol allowed dose reduction. The treating physician decided whether bevacizumab (7.5 mg/kg intravenously, q3w) was added or not. Response evaluation was conducted after every three cycles (every nine weeks) and evaluated according to the Response Evaluation Criteria In Solid Tumors version 1.1 [45]. Patients were treated until progression, unacceptable toxicity, or patients’ wish for a treatment break.

### 2.3. Biomarker Analysis

Baseline blood sampling was conducted at the time of the inclusion. Blood samples were centrifuged at 4 °C at 2300× *g* for 10 min and serum was aliquoted and frozen at minus 80 °C until the analysis of YKL-40 and IL-6. Routine hematological (ANC and WBC) and CRP analysis were conducted at the local laboratories, while the analysis of YKL-40 and IL-6 were determined at Department of Oncology and Medicine, Copenhagen University Hospital—Herlev and Gentofte.

The cut-off value for CRP was defined as 10 mg/L in accordance to the internationally accepted elevated level. The cut-off value for dNLR was defined as 2.22 [29].

YKL-40 was measured using an enzyme-linked immunosorbent assay (ELISA) (Quidel Corporation, San Diego, CA, USA) in accordance with the manufacturer’s instructions. The lower limit of detection for YKL-40 was 20 µg/L, the intra- and inter-assay coefficients of variation (CVs) were <5% and <6%, respectively. The cut-off value for YKL-40 was defined as 200 µg/L (the 95th percentile value in older healthy subjects) [32].

IL-6 was measured using a high sensitive ELISA (Quantikine HS600B, R&D Systems, Abingdon, UK) in accordance with the manufacturer’s guidance. The lower limit of detection was 0.01 ng/L, and the intra- and inter-assay CVs were 8% and 11%, respectively. The cut-off value for IL-6 was defined as 4.5 ng/L in accordance to the international standard [46].

The measurement of CRP, YKL-40, IL-6, and the calculation of dNLR were conducted blinded to patient characteristics and study outcomes according to the REMARK recommendations.

### 2.4. Covariates

The demographic and clinical characteristics of the study participants were obtained by review of the electronic medical records blinded to the results of the biomarker analysis and calculation of dNLR. We identified and decided to include the following variables affecting the clinical outcomes: age, sex, the Eastern Cooperative Oncology Group performance status (ECOG PS), and treatment allocation.

### 2.5. Statistics

For baseline demographic and clinical characteristics, we applied descriptive statistics. Data were presented as median value (interquartile range) or n (%), as appropriate. To visualize the age distribution of included patients recommended by the REMARK guideline, a histogram was created. For variables, not showing normal distribution, log transformation was applied. Depending on the number of observations, for categorical binary variables chi2-test or Fischer’s exact test was used, for continuous numerical variables the Wilcoxon Mann-Whitney test was applied.

### 2.6. Survival Analyses

Outcomes were defined as OS and PFS; survival curves were estimated by the Kaplan-Meier method. The date of follow-up was the 1 September 2018.

The comparisons between subgroups were performed by log-rank test. Hazard ratios (HR) and corresponding 95% confidence intervals (95%CI) were estimated by Cox proportional hazard regression; the proportional hazards assumptions were tested based on Schoenfield residuals.

### 2.7. Multivariable Analyses

Multivariate Cox proportional hazards regression models were applied for the survival outcomes. The relevant clinical covariates were included; the model hence was fitted to age, sex, ECOG PS, treatment allocation, number of metastatic sites, and primary tumor in situ. To avoid that the model would be over-fitted, we balanced the number of co-variables according to the number of observations. Being able to demonstrate and compare the prognostic value of the biomarkers, we applied C-statistics and calculated Harrell’s C (area under the curve) with 95% CIs. Two-sided *p*-values ≤ 0.05 were considered statistically significant and estimates were reported with 95%CI. We performed data analysis in STATA v17 (StataCorp LLC, College Station, TX, USA).

### 2.8. Missing Data

In the dataset, 2–19% of observations were missing for the variables of interest, due to three centers chose not to participate in sampling of blood for biomarker analysis and/or procedural error. We concluded that those observations were reasonable to exclude from the analyses.

### 2.9. Sample Size

The intention-to-treat population (consisting of all randomized patients) was 160 patients. No formal sample size calculation was performed for this biomarker analysis.

## 3. Results

### 3.1. Patient Population

One hundred and sixty patients were included in the NORDIC9-study between March 2015 and October 2017 of whom 157 were available for biomarker analysis. The inclusion was stopped when the required number of patients was obtained. The median follow-up was 23.8 months (interquartile range (IQR): 18.8–30.9). The patient flow is presented by a CONSORT 2010 diagram (Figure 1).

Demographic and clinical characteristics were balanced between the treatment arms (Table 1). The median age in the ITT population was 78 years (IQR: 76–81). Age distribution is illustrated as a histogram in accordance to the REMARK guideline (Appendix A).

The pre-treatment levels of the inflammatory biomarkers and their association to demographic and baseline clinical covariates in the ITT population are shown in Table 2. CRP, dNLR, YKL-40, and IL-6 levels were significantly elevated in patients presented with either weight loss, metastatic sites ≥3, primary tumor in situ, or ECOG PS 2 (except YKL-40). CRP, YKL-40, and IL-6 were also significantly higher in patients with elevated lactate dehydrogenase, alkaline phosphatase, and carcinoembryonic antigen (Table 2).

### 3.2. Univariate Analyses

Kaplan-Meier plots for each biomarker and OS and PFS are illustrated in Figure 2 and Figure 3, respectively.

#### 3.2.1. OS

Largest difference in OS was seen favoring the patients with normal CRP: 21.9 vs. 8.7 months, (95%CI: (20.1–27.4) vs. (6.1–10.6), HR = 3.36 (95%CI: 2.23–5.08), *p* < 0.001) (Figure 2, Appendix A). Patients with dNLR ≤2.2 had significantly longer OS than those with dNLR >2.2: 19.3 vs. 10.3 months (95%CI: (14.4–21.4) vs. (6.6–12.6), HR = 1.89 (95%CI: 1.29–2.75), *p* = 0.001). Elevated YKL-40 was associated with statistically significant shorter OS: 19.5 vs. 10.4 months (95%CI: (13.1–21.9) vs. (5.7–13.9), HR = 1.81, (95%CI: 1.20–2.74), *p* = 0.005), respectively (Figure 2, Appendix A). In patients with normal IL-6, OS tended to be longer (20.8 vs. 11.5 months), though statistical significance was not reached (HR = 1.55 (95%CI: 0.99–2.42), *p* = 0.053).

The patients who were randomized to receive full-dose S1, had a significantly higher pre-treatment CRP compared to patients treated with reduced-dose SOx (19.0 vs. 9.0 mg/L, *p* = 0.034) (Table 2). Therefore, we conducted an analysis stratifying patients by treatment arms (S1 vs. SOx) and CRP levels (CRP high vs. CRP low) (Appendix A). Regarding OS, the Kaplan-Meier curves were clearly separated by CRP levels regardless of treatment arms. Appendix A demonstrates significant prolonged OS for patients with low CRP levels, irrespective of the received treatment.

#### 3.2.2. PFS

Patients with normal CRP had PFS 8.2 vs. 4.1 months (95%CI: (6.6–9.1) vs. (3.4–4.8), HR = 1.85 (95%CI: 1.32–2.58), *p* < 0.001) compared to those with elevated values (Figure 3, Appendix A). The PFS was 6.8 vs. 4.2 months in favor of the dNLR ≤ 2.2 sub-group (95%CI: (5.9–8.3) vs. (3.7–6.2), HR = 1.42 (95%CI: 1.03–1.96), *p* = 0.034). YKL-40 did not show statistical significant difference regarding PFS, while IL-6 did: 8.2 vs. 4.7 months (95%CI: (5.9–8.9) vs. (4.1 vs. 6.2), HR = 1.52 (95%CI: 1.04–2.21), *p* = 0.030) in favor of patients with IL-6 <4.5 pg/L. (Figure 3, Appendix A).

### 3.3. Multivariable Analyses

#### 3.3.1. OS

Both CRP, dNLR, YKL-40, and IL-6 demonstrated statistically significant differences in HRs, with the highest HR observed for CRP: HR = 3.40 (95%CI: 2.20–5.26), *p* < 0.001) (Table 3).

#### 3.3.2. PFS

Statistically significant differences between sub-groups were found for CRP and IL-6, but not for dNLR and YKL-40 (Table 3). The highest HR was observed in patients with elevated CRP (HR = 1.64 (95%CI: 1.16–2.34), *p* = 0.005).

#### 3.3.3. C-Statistics

A good prognostic value of CRP was established in our cohort regarding OS with a Harrell’s C value at 0.72 (95%CI: 0.67–0.76) (Table 3). Models including YKL-40, IL-6, and dNLR provided moderate prognostic value.

## 4. Discussion

In this RCT using data from older vulnerable patients with mCRC treated with palliative chemotherapy, high pre-treatment CRP and IL-6 were independently associated to short OS and PFS. High dNLR and YKL-40 was also independently associated with short OS, but not to PFS. When C-statistics was applied, CRP provided a good prognostic model, while dNLR, YKL-40, and IL-6 demonstrated a moderate value.

### 4.1. Explanation, Interpretation

All selected pre-treatment circulating biomarkers demonstrated clinical value in this homogenous prospective cohort of vulnerable older patients with mCRC, thus may be considered as useful biomarkers in daily clinical practice providing information about life expectancy, facilitating discussion on prognosis and supporting decision-making.

Given the sharply increasing number of older adults with cancer, there is an unmet need for being able to identify those who will not likely benefit of anti-neoplastic treatment. GA is still not implemented at most oncology departments, whereas prognostic biomarkers including circulating tumor cells (cTCs) or even circulating tumor deoxyribonucleic acid (ctDNA) and predictive biomarkers like *RAS* mutation or mismatch repair status play an emerging role in daily practice [47,48,49,50]. To understand, what biomarkers tell us about the clinical outcomes, and how these biomarkers should be optimally used, we need to define clearly, what prognostic and predictive value means. In essence, both are used in the context to foresee a possible outcome for the patient with a specific disease. A prognostic factor is an objectively measurable clinical or biologic characteristic telling us about the likely outcome of the cancer in untreated individuals [51,52]. In contrast, a predictive factor is a clinical or biologic characteristic that identifies sub-groups of treated patients having different outcomes as a consequence of the treatment, thus, it tells as about the benefit of the treatment [51,52].

Despite these definitions seem to be straightforward and easy to understand, the terms prognostic and predictive are often imprecisely used in the literature causing confusion [51,52]. On the one hand, some biomarkers are both prognostic and predictive, like the presence of *BRAF*-mutation in mCRC. On the other hand, some factors e.g., circulating biomarkers are associated with the cancer and its course strong enough to foresee an outcome, without being the direct cause of the outcome, or being able to predict whether the treatment will work in a subgroup or not. Therefore, a predictive factor can foresee the modifying effect of the treatment on the course of the disease, and this can only be appropriately investigated by RCTs, for each specific treatment regimen [51]. 

In the present study, we tested the prognostic value of four inflammatory biomarkers in older vulnerable patients treated with either monotherapy S1 or reduced dose of combination chemotherapy S1 and oxaliplatin. However, the NORDIC9-study was not designed to investigate the predictive value regarding the effect of palliative chemotherapy.

### 4.2. Comparison to Other Studies

CRP, dNLR, YKL-40, and IL-6 are biomarkers of systemic inflammation and elevated plasma levels are frequently seen in several age-related conditions, like cancer, and associated with the presence of vulnerability and frailty in older adults [12,17,22,26,37,38,39,40,41,42].

A study investigating post-adjuvant circulating biomarkers in patients with CRC found that elevated CRP and IL-6 were associated with shorter disease-free survival [18]. Despite the primary tumor was surgically removed and the patient received adjuvant chemotherapy, the elevated levels of CRP and IL-6 were associated with poor prognosis. In the metastatic setting, elevated pre-treatment CRP and IL-6 were independently associated with shorter PFS and OS, and provided useful information on prognosis in addition to the tumor mutational status (*RAS* and *BRAF*) [21]. The same trend can be observed in patients with metastatic solid tumors as well, broadening the applicability of CRP and IL-6 to other primary malignancies [15].

Several studies have demonstrated that elevated dNLR and NLR are associated with worse survival outcomes in patients with solid tumors, both in early stage and in the metastatic setting [29,30]. The MRC COIN study included 1630 patients with mCRC and applied the same cut-off for dNLR (2.22) and found a HR (HR = 1.70, 95%CI: 1.52–1.90, *p* < 0.001) for OS between the sub-groups similar to HR values in our cohort in both uni- and multivariable analyses [29]. The prognostic role of NLR using a cut-off at 3 was also confirmed in the TRIBE study [53]. The NLR was a strong prognostic biomarker for OS in both uni- and multivariable analyses. However, they found no difference between the sub-groups regarding PFS, emphasizing that pre-treatment NLR did not affect treatment efficacy, thus, cannot be considered as a predictive biomarker.

The prognostic role of YKL-40 is well established and broadly investigated through different types of malignancies [33]. In patients with mCRC treated with oxaliplatin-based doublet, the prognostic role of elevated YKL-40 was confirmed and resulted in significant shorter OS [36]. These findings are in line with our results.

### 4.3. Methodological Considerations—Strengths and Limitations

Among the strengths of the NORDIC9-study biomarker analysis was that we have pre-specified the analysis in the protocol and prospectively collected the data in a homogenous cohort, in terms of both age and diagnosis. Moreover, 157 of 160 included patients (94%) were available for biomarker analysis. Most of studies investigating these biomarkers were conducted in highly selected fit and younger patients without significant comorbidities [53], however, in contrast to these studies, our patients represent those clinicians usually treat in daily clinical practice. Thus, our results may guide treatment decisions in majority of patients with mCRC. As we demonstrated above, the statistical difference in baseline CRP levels between treatment arms did not pose a selection bias.

In addition, it is important to note, that YKL-40 and IL-6 were analyzed centrally reducing possible bias due to different methods.

The study however has also some limitations. Up to 19% of YKL-40 and IL-6 samples were missing. CRP together with ANC and WBC were measured in the local laboratories at the participating sites, and this may cause some heterogeneity in the data due to different methods. 

### 4.4. Implication for Clinical Practice

CRP and dNLR are easily available, affordable, and may be routinely used adding valuable information on survival outcomes in older patients with mCRC. YKL-40 and IL-6 are not used routinely in the clinical practice and did not provide additional information regarding prognosis.

## 5. Conclusions

CRP, dNLR, YKL-40, and IL-6 demonstrated prognostic value in vulnerable older patients with mCRC receiving palliative chemotherapy. CRP is easily available, low-cost, and a reliable prognostic biomarker for daily clinical practice and may add important information when prognosis and treatment options are discussed with patients and caregivers, thus may enhance shared decision-making.

## Figures and Tables

**Figure 1 jcm-11-05603-f001:**
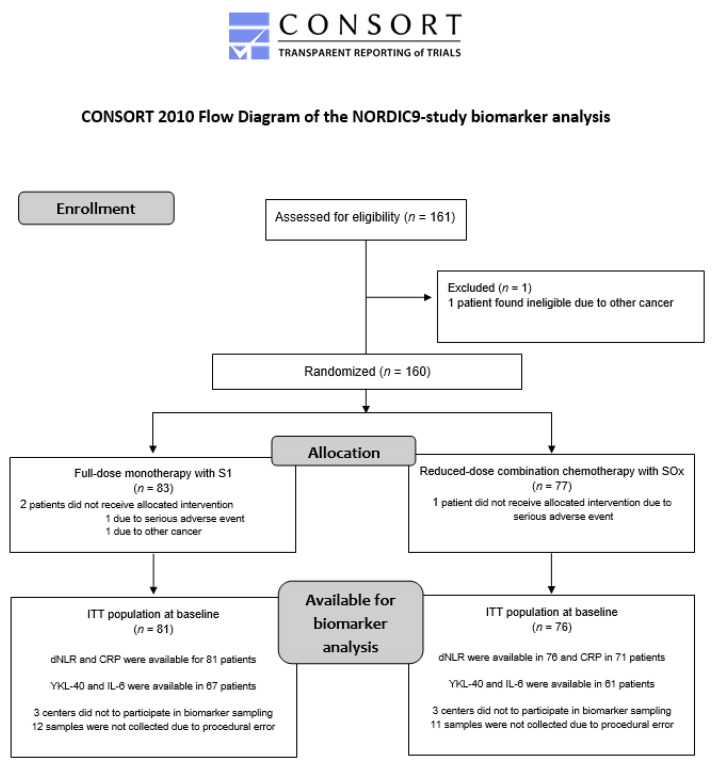
CONSORT 2010 diagram presenting patient flow in the NORDIC9-study. Abbreviations: ITT: intention-to-treat, dNLR: derived neutrophil-to-lymphocyte ratio, CRP: C-reactive protein.

**Figure 2 jcm-11-05603-f002:**
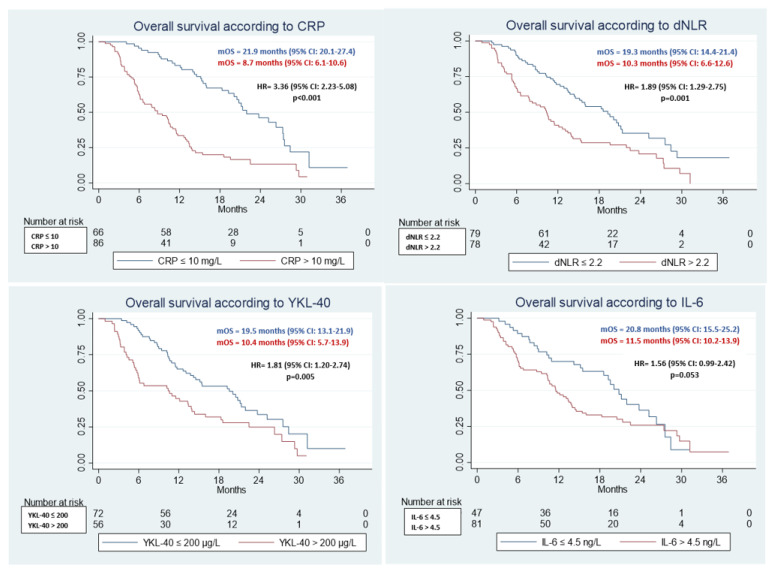
Kaplan-Meier plots showing baseline dichotomized biomarker (normal/high) associations with overall survival.

**Figure 3 jcm-11-05603-f003:**
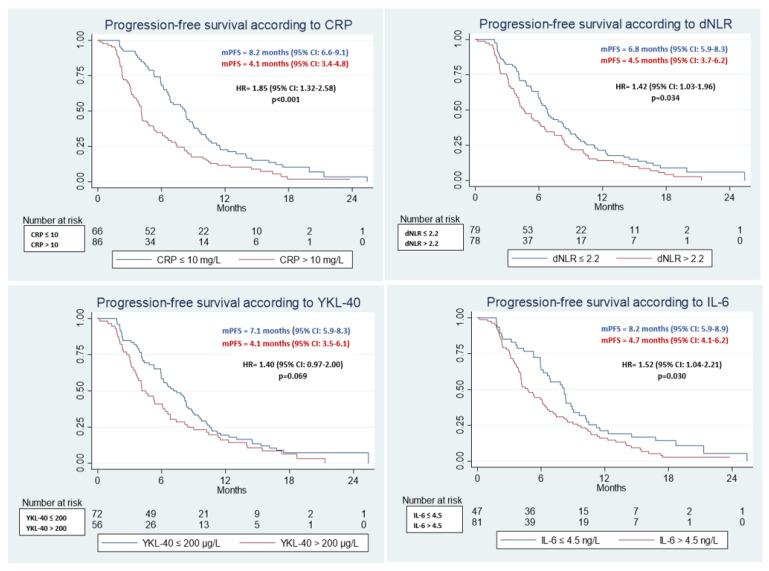
Kaplan-Meier plots showing baseline dichotomized biomarker (normal/high) associations with progression-free survival.

**Table 1 jcm-11-05603-t001:** Demographic and baseline clinical characteristics of NORDIC9-study population.

Demographic and Baseline Clinical Characteristics Data Presented as Median (Interquartile Range) or n (%)	NORDIC9-Study Population Available for Biomarker Analysis *n* = 157	NORDIC9-Study Treatment Arms	
		Full-Dose Monotherapy Arm A *n* = 81	Reduced-Dose Combination CT Arm B *n* = 76
**Age**			
**Median age** in years (IQR)	78 (75–81)	78 (76–81)	77 (75–80)
**Sex**			
Female	78 (50%)	40 (49%)	38 (50%)
Male	79 (50%)	41 (51%)	38 (50%)
**ECOG Performance status**			
0	52 (33%)	29 (36%)	23 (30%)
1	75 (48%)	37 (46%)	38 (50%)
2	30 (19%)	15 (18%)	15 (20%)
**Surgery for primary tumor**			
No	68 (43%)	36 (44%)	32 (42%)
Yes	89 (57%)	45 (56%)	44 (58%)
**Prior adjuvant chemotherapy**			
Yes	29 (18%)	18 (22%)	11 (14%)
No	128 (82%)	63 (78%)	65 (86%)
**Number of metastatic sites**			
1–2	95 (61%)	47 (58%)	38 (63%)
≥3	62 (39%)	34 (42%)	28 (37%)
**Self-reported weight loss > 5% within the last 2 months**			
No	122 (78%)	58 (72%)	64 (84%)
Yes	35 (22%)	23 (28%)	12 (16%)
***RAS* and *BRAF* status**			
*RAS* and *BRAF* wild-type	36 (23%)	21 (26%)	15 (20%)
*RAS* or *BRAF* mutated	78 (50%)	39 (48%)	39 (51%)
Unknown	43 (27%)	21 (26%)	22 (29%)
**Lactate dehydrogenase (U/L)**			
≤255	93 (59%)	44 (54%)	49 (64%)
>255	54 (34%)	29 (36%)	25 (33%)
Unknown	10 (7%)	8 (10%)	2 (3%)
**Alkaline phosphatase (U/L)**			
≤105	90 (57%)	49 (60%)	41 (54%)
>105	66 (42%)	32 (40%)	34 (45%)
Unknown	2 (1%)	0 (0%)	1 (1%)
**White blood cells (10^9^/L)**			
≤10	113 (72%)	56 (69%)	57 (75%)
>10	44 (28%)	25 (31%)	19 (25%)
**Neutrophil granulocytes (10^9^/L)**			
≤8	127 (81%)	65 (80%)	62 (82%)
>8	30 (19%)	16 (20%)	14 (18%)
**C-reactive protein (mg/L)**			
≤10	65 (41%)	27 (33%)	38 (50%)
>10	84 (54%)	53 (65%)	32 (42%)
Unknown	8 (5%)	2 (2%)	6 (8%)
**Derived neutrophil-to-lymphocyte ratio**			
≤2.2	79 (50%)	41 (50%)	38 (50%)
>2.2	78 (50%)	40 (50%)	38 (50%)
**YKL-40 (µg/L)**			
≤200	70 (45%)	32 (40%)	38 (50%)
>200	56 (36%)	34 (42%)	22 (29%)
Unknown	31 (19%)	15 (18%)	16 (21%)
**Interleukin-6 (ng/L)**			
≤4.5	48 (31%)	21 (26%)	27 (36%)
>4.5	78 (50%)	45 (56%)	33 (43%)
Unknown	31 (19%)	15 (18%))	16 (21%)
**Carcinoembryonic antigen (µg/L)**			
≤5	30 (19%)	12 (15%)	18 (24%)
>5	121 (77%)	65 (80%)	56 (74%)
Unknown	6 (4%)	4 (5%)	2 (2%)

**Table 2 jcm-11-05603-t002:** Demographic and baseline clinical characteristics in the intention-to-treat population in the NORDIC9-study and their association with pre-treatment plasma CRP, derived neutrophil-lymphocyte ratio, YKL-40, and IL-6.

Baseline Demographic and Clinical Characteristics		n* (%)	CRP Median (mg/L) (IQR)	*p*-Value	n* (%)	dNLR Median (IQR)	*p*-Value	n* (%)	YKL-40 Median (µg/L) (IQR)	*p*-Value	n* (%)	IL-6 Median (ng/L) (IQR)	*p*-Value
**Sex**	Male	78	12 (5–36)	0.889	79	2.2 (1.75–2.98)	0.416	68	179 (116–323)	0.371	68	6.9 (3.6–17.8)	0.463
	Female	74	17 (5–43)		78	2.2 (1.6–2.9)		60	168 (109–268)		60	7.0 (2.6–14.2)	
**ECOG PS**	0–1	122	11 (5–33)	**0.004**	127	2.1 (1.6–2.6)	**0.004**	101	159 (110–276)	0.069	101	6.0 (2.9–12.7)	**0.001**
	2	30	29 (10–80)		30	2.9 (2.0–3.3)		27	224 (143–402)		27	18.1 (4.2–40.2)	
**Treatment arm**	S1	81	19 (6–48)	**0.034**	81	2.2 (1.6–2.9)	0.864	67	204 (118–301)	0.440	67	7.1 (3.7–17.4)	0.469
	SOx	71	9 (5–33)		76	2.2 (1.7–2.9)		61	163 (98–282)		61	6.6 (2.9–14.7)	
**Resection of primary tumor**	Yes	88	10 (4–28)	**0.001**	89	2.0 (1.5–2.5)	**0.003**	76	144 (95–273)	**0.005**	76	5.0 (2.4–13.8)	**0.013**
	No	64	25 (8–55)		68	2.5 (1.8–3.0)		52	224 (156–318)		52	9.3 (4.1–23.7)	
**Adjuvant chemotherapy**	Yes	29	6.0 (4–19)	**0.026**	29	2.0 (1.4–2.5)	0.058	26	133 (80–273)	0.196	26	4.9 (2.1–9.4)	**0.022**
	No	123	17 (5–48)		128	2.3 (1.7–3.0)		102	177 (118–296)		102	8.3 (3.4–18.2)	
**Number of metastatic sites**	≤2	92	10.0 (4–28)	**0.001**	95	2.0 (1.5–2.6)	**0.016**	79	145 (97–263)	**0.003**	79	5.6 (2.8–13.6)	**0.036**
	≥3	60	24 (7–57)		62	2.4 (1.8–3.1)		49	224 (154–345)		49	9.8 (3.9–23.7)	
**Weight loss > 5% in the last 2 months**	No	116	11.0 (4–30)	**0.001**	122	2.2 (1.6–2.8)	**0.037**	97	157 (100–272)	**0.006**	97	5.6 (2.7–13.0)	**0.005**
	Yes	36	32 (12–86)		35	2.5 (2.0–4.0)		31	245 (133–556)		31	15.1 (5.6–34.9)	
***RAS* and *BRAF* mutation status**	*RAS* and *BRAF* wild type	34	10.5 (5–35)	0.866	36	2.2 (1.6–2.6)	0.846	33	204 (143–272)	0.194	33	8.7 (3.4–14.7)	0.881
	*RAS* or *BRAF* mutated	76	14.5 (5–44)		77	2.1 (1.7–2.9)		62	157 (95–296)		62	6.8 (3.8–14.5)	
**Carcinoembryonic-antigen (µg/L)**	≤5	29	5 (4–17)	**0.012**	30	1.9 (1.5–2.3)	0.043	25	149 (80–204)	**0.026**	25	5.0 (2.9–8.7)	**0.029**
	>5	117	17 (6–45)		121	2.3 (1.7–2.9)		98	189 (119–301)		98	7.9 (3.4–21.7)	
**Alkaline-phosphatase (U/L)**	≤105	85	7 (4–21)	**<0.001**	90	2.1 (1.6–2.8)	0.103	74	139 (95–212)	**<0.001**	74	4.6 (2.5–12.2)	**0.002**
	>105	67	28 (14–59)		67	2.4 (1.8–3.1)		54	262 (154–556)		54	12.1 (5.1–30.6)	
**Lactate-dehydrogenase (U/L)**	≤255	90	10 (5–27)	**0.001**	93	2.1 (1.6–2.8)	0.073	80	154 (87–258)	**0.009**	80	5.0 (2.6–11.8)	**0.001**
	>255	62	27 (7–59)		64	2.4 (1.8–3.1)		48	216 (142–363)		48	12.8 (4.9–27.5)	
**C-reactive protein (mg/L)**	≤10	NA	NA	NA	65	1.8 (1.5–2.3)	**<0.001**	56	119 (80–179)	**<0.001**	56	3.3 (2.1–5.4)	**<0.001**
	>10				84	2.5 (2.0–3.3)		66	248 (155–481)		66	13.9 (7.5–29.5)	

Cut-off values for CRP = 10 mg/L, dNLR = 2.2, YKL-40 = 200 µg/L, and IL-6 = 4.5 ng/L. Abbreviations: CRP, C-reactive protein; IQR, inter quartile range; ECOG PS, Eastern Cooperative Oncology Group Performance Status; ITT, Intention to treat; IL-6, Interleukin-6; and dNLR, derived neutrophil-to-lymphocyte ratio, NA: non-applicable. n*: Values may be different due to not every center participated in the biomarker sub-study (plasma YKL-40 and IL-6) and missing values.

**Table 3 jcm-11-05603-t003:** Multivariate Cox analysis of inflammatory biomarkers and progression-free survival and overall survival adjusted for age, sex, treatment allocation, ECOG performance status, number of metastatic sites, and primary tumor in situ in the NORDIC9-study.

Biomarker	n	Progression-Free Survival			Overall Survival		
		Hazard Ratio (95%CI)	*p*-Value	Harrell’s C (95%CI)	Hazard Ratio (95%CI)	*p*-Value	Harrell’s C (95%CI)
**CRP** (mg/L)							
≤10	66	1.00	**0.005**	0.66 (0.61–0.70)	1.00	**<0.001**	0.72 (0.67–0.76)
>10	86	**1.65** (1.16–2.34)			**3.40** (2.20–5.26)		
**dNLR**							
≤2.2	79	1.00	0.088	0.63 (0.58–0.68)	1.00	**0.013**	0.66 (0.60–0.71)
>2.2	78	**1.34** (0.96–1.88)			**1.65** (1.11–2.44)		
**YKL-40** (µg/L)							
≤200	72	1.00	0.250	0.63 (0.59–0.68)	1.00	**0.017**	0.66 (0.60–0.73)
>200	56	1.26 (0.851.87)			**1.71** (1.10–2.66)		
**IL-6** (ng/L)							
≤4.5	47	1.00	**0.024**	0.64 (0.59–0.69)	1.00	**0.044**	0.65 (0.59–0.71)
>4.5	81	**1.58** (1.06–2.35)			**1.60** (1.02–2.52)		

## Data Availability

The data that support the findings of this study are available on request from the corresponding author (G.L.).

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
