# Peer review of "The Prognostic Value of Pre-Treatment Circulating Biomarkers of Systemic Inflammation (CRP, dNLR, YKL-40, and IL-6) in Vulnerable Older Patients with Metastatic Colorectal Cancer Receiving Palliative Chemotherapy—The Randomized NORDIC9-Study"

_jcm, 2022, doi:10.3390/jcm11195603_

Round 1
Reviewer 1 Report
This is a very well written manuscript. I have no comments in relation to the choice of biomarkers or the design of the study.
I am just a little sceptical about 2 points raised below, which the authors may want to comment on:
1. The investigators chose not to use biological agents (ant-VEGF, anti-EGFR) in NORDIC-9. The outcomes in the high CRP (vs low) and low dNLR (vs high) patients in terms of OS are substantially worse. One wonders whether other treatments such as capecitabine/ bevacizumab (AVEX) or 5FU/ panitumumab (PANDA) could have affected the prognostic value of the biomarkers used?
2. Following from 1, it's not entirely clear how would the results of such biomarker analysis help in the shared decision making as suggested in the concluding remarks. For example, would high CRP patients be asked to think twice before proceeding with systemic therapy? Would they be offered biological agents as well? Perhaps you may want to elaborate.
Author Response
Please, see the attached document. Thank you.

Reviewer 2 Report
Overall I really like this manuscript, it is an interesting topic in an important and growing population. I have some suggestion for the authors.
Most important for me is that you are talking about vulnerable patients. But I have no idea how you define vulnerable. I cannot conclude it from your baseline table. At the same time the NORDIC9 trial included GA parameters. In my opinion you make this manuscript much stronger by adding GA details and also looking at the association with the biomarkers and the GA. Assuming that vulnerable/frail patients have higher inflammatory markers and worse outcomes makes it necessary to look into it. Besides that, it make your study more unique.
Please correct the multivariate analysis also for metastatic site > 3 and primary tumor in situ in the first analysis. I think that makes more sense than doing stepwise multivariate models after your multivariate analysis.
Please combine table 3 and 4 (or table 3 supplementary). Table 5 is confusing and too big. Maybe just remove it or replace it to supplementary.
I would be interested in stratification on treatment arm and see if CRP has the same effect in both? (despite already adjustment for treatment allocation). Full dose S1 had higher CRP levels. And the full dose S1 group was also the one with the worse prognosis.
Part of the biomarker measurements was done in the participating hospital itself, and partly centrally. Measurements for NLR and CRP can give differences in measurements in different centres. Please mention as a limitation (now only strength that other biomarkers are centrally measured)
Missing data (line 200): st2-15% missing but YKL-40 and IL-6 misses 19% of (table 1) adjust and also name in limitations
Please use the same terminology. It now confusing (e.g. line 275-278): diagnostic ability, prognostic value. In abstract, correlation... and in the discussion again about prognostic and predictive.
Author Response

(The authors gave the same response as above.)
